# A Tailored Strategy to Crosslink the Aspartate Transcarbamoylase Domain of the Multienzymatic Protein CAD

**DOI:** 10.3390/molecules28020660

**Published:** 2023-01-09

**Authors:** Francisco del Caño-Ochoa, Antonio Rubio-del-Campo, Santiago Ramón-Maiques

**Affiliations:** 1Instituto de Biomedicina de Valencia (IBV), CSIC, Jaime Roig 11, 46010 Valencia, Spain; 2Group CB06/07/0077 at the Instituto de Biomedicina de Valencia (IBV-CSIC) of CIBERER-ISCIII, Centro de Investigación Biomédica en Red de Enfermedades Raras, Melchor Fernández Almagro 3, 28029 Madrid, Spain

**Keywords:** nucleotide metabolism, de novo pyrimidine biosynthesis, carbamoyl-phosphate synthetase, dihydroorotase, cysteine, disulfide bridge, protein stability, X-ray crystallography

## Abstract

CAD is a 1.5 MDa hexameric protein with four enzymatic domains responsible for initiating de novo biosynthesis of pyrimidines nucleotides: glutaminase, carbamoyl phosphate synthetase, aspartate transcarbamoylase (ATC), and dihydroorotase. Despite its central metabolic role and implication in cancer and other diseases, our understanding of CAD is poor, and structural characterization has been frustrated by its large size and sensitivity to proteolytic cleavage. Recently, we succeeded in isolating intact CAD-like particles from the fungus *Chaetomium thermophilum* with high yield and purity, but their study by cryo-electron microscopy is hampered by the dissociation of the complex during sample grid preparation. Here we devised a specific crosslinking strategy to enhance the stability of this mega-enzyme. Based on the structure of the isolated *C. thermophilum* ATC domain, we inserted by site-directed mutagenesis two cysteines at specific locations that favored the formation of disulfide bridges and covalent oligomers. We further proved that this covalent linkage increases the stability of the ATC domain without damaging the structure or enzymatic activity. Thus, we propose that this cysteine crosslinking is a suitable strategy to strengthen the contacts between subunits in the CAD particle and facilitate its structural characterization.

## 1. Introduction

Pyrimidine nucleotides are essential components of DNA and RNA and key activator molecules for protein glycosylation and lipid synthesis. The pathway for de novo pyrimidine biosynthesis consists of six conserved sequential steps to produce uridine 5′-monophosphate (UMP) (Figure 1a), but the organization of the pathway-initiating enzymes differs greatly from bacteria to humans [1,2]. Thus, whereas in prokaryotes these enzymes are individually encoded, in animals, glutaminase (GLN)-dependent carbamoyl phosphate synthetase (CPS-2), aspartate transcarbamoylase (ATC), and dihydroorotase (DHO) are fused as functional domains within a single polypeptide of ~2253 amino acids (aa) named CAD (Figure 1a) [2,3]. Moreover, in fungi, GLN, CPS, and ATC are fused into a CAD-like protein with an inactive DHO-like domain, and the DHO activity is encoded as an independent enzyme [4,5].

CAD initiates and controls the flux through the pathway and is regulated by allosteric effectors and phosphorylations through different signaling cascades [6,7,8,9]. Therefore, the activities of this multienzymatic protein are key for normal cell growth and proliferation, and its alterations are of biomedical importance. Indeed, CAD is found upregulated in tumors [10,11,12], while mutations that compromise its function are the cause of a severe infantile epileptic disease (OMIM #616457) [13,14,15,16].

Since its discovery, CAD has been known to self-assemble into a mixture of oligomers, mainly hexamers of ~1.5 MDa [3,17]. However, attempts to explain the architecture and functioning mechanisms of this megaenzyme have been hampered by the large size of the protein and the high sensitivity of interdomain regions to proteolytic cleavage [18,19,20]. Consequently, knowledge about CAD is scarce and fragmented, as the only detailed structural information available is for the isolated ATC and DHO domains [2]. The crystal structures of the ATC domains of human CAD and the CAD-like protein from the fungus *Chaetomium thermophilum* showed a dome-shaped homotrimer with three active sites between the subunits that are very similar to bacterial and plant ATCs [18,21,22,23]. On the other hand, the crystal structure of the DHO domain of human CAD showed a homodimer with one active center per subunit containing three Zn^2+^ ions coordinated by a carboxylated lysine and a negatively charged histidine [24,25]. This active site is missing in the *C. thermophilum* DHO-like domain. Still, the crystal structure of the inactive DHO-like domain showed a very similar homodimer to human DHO, supporting that regardless of enzymatic activity, DHO may play a structural role in the architecture of CAD [18].

Based on the assembly of ATC and DHO as trimers and dimers, respectively, a model for the architecture of CAD was proposed in which trimers of the protein, hooked by their ATC domains, could dimerize through the DHO domains, thus forming a hexamer or “dimer of trimers” (Figure 1b) [7,18]. In support of the model, mutations that disrupt ATC trimerization or DHO dimerization were shown to cause the dissociation of CAD hexamers into dimers and trimers, respectively [18,26]. Therefore, the DHO and ATC domains could form a central scaffold surrounded by GLN and CPS-2 domains with a predicted structural resemblance (https://alphafold.ebi.ac.uk/entry/P27708; accessed on 14 November 2022) to *E. coli* CPS [27] and human CPS-1 [28]. Interestingly, both *E. coli* CPS and human CPS-1 form homodimers in a solution; thus, the dimerization of CPS-2 could also contribute to the assembly (Figure 1b). However, despite the attractiveness of the model, the reconstruction of the mega-enzyme is still pending.

Recently, we obtained recombinant full-length *C. thermophilum* CAD-like hexamers in high yield and purity, thus avoiding mammalian CAD degradation problems (unpublished results). However, the CAD-like particles fell apart during the preparation of grids for structural characterization by cryo-electron microscopy (EM). Previously, we experienced similar problems with shorter constructs spanning the DHO and ATC domains of human CAD and *C. thermophilum* CAD-like proteins [18]. These truncated constructs were assembled as hexamers in solution, but the preparation of diluted samples for EM promoted the dissociation of the DHO-ATC particles, increased the heterogeneity, and prevented the 3D reconstruction of the complex. Therefore, additional steps are needed to improve the stability of the protein particles. Here, we report the insertion by site-directed mutagenesis of two cysteine (Cys) residues in the ATC domain that favor the formation of disulfide bridges between subunits in the trimer. Combining size-exclusion chromatography (SEC) and X-ray crystallography with thermostability and enzymatic assays, we proved that the covalent linkages increase the stability of the ATC trimer without damaging the structure or enzyme activity. We propose this specific crosslinking as a suitable strategy to strengthen the contacts in the CAD particle and, thus, facilitate its structural characterization.

## 2. Results

### 2.1. Mutant Production

We explored the crystal structure of the ATC domain of *C. thermophilum* CAD-like protein (ctATC; PDB ID 5NNN) to identify positions that could be replaced by Cys and favor the formation of intersubunit disulfide bridges. The candidate residues should be ~4 Å apart for covalent linkage, distant from the active site or the moving catalytic loops to preserve enzymatic activity, and not in hydrophobic patches nor interacting with other residues that could affect protein folding. Using these criteria, we selected residue N2045, in helix α3, and residue R2238, in helix α12 (Figure 2a,b). Interestingly, the N2045 position is occupied by a Cys in most of the analyzed CAD-like sequences, as well as in all mammalian CAD proteins, and therefore, mutation N2045C was expected to have a neutral effect on ctATC (Figure 2a). In contrast, the R2338 position is occupied by either Arg or Lys in fungal CAD-like proteins or by Glu in mammalian CAD (Figure 2a,b), so the effect of a change to Cys was uncertain. 

We previously reported the structure of ctATC, but the crystals appeared by serendipity upon cleavage of a larger protein spanning the ctATC and DHO-like domains [18]. So, for this study, we engineered a construct encoding only the ctATC domain (ctATC-WT; aa 1939–2253) fused to a removable N-terminal His_6_-maltose binding protein (MBP) tag and then introduced the mutations N2045C and R2238C (ctATC2Cys). Both ctATC-WT and ctATC2Cys were produced in similar yields and purity (Appendix A), indicating that the mutations did not impair the folding or decrease the solubility of the protein. Analysis of purified ctATC2Cys by SDS-PAGE under non-reducing conditions confirmed the formation of crosslinked dimers and trimers in solution (Figure 2c), and size-exclusion chromatography (SEC) profile showed that, similar to ctATC-WT, the mutated protein elutes in a single peak with an estimated molecular weight of 119 kDa, as expected for a homotrimer (Figure 2d).

### 2.2. Crystal Structure of the Crosslinked ctATC2Cys Trimer

Purified ctATC2Cys formed bipyramidal-shaped crystals in less than 20 min under conditions similar to those previously reported [18] (Appendix A). Crystals diffracted X-rays to 1.5 Å resolution and belonged to cubic space group P4_3_32 (Table 1). Structural determination by molecular replacement showed one protein subunit per asymmetric unit forming a trimer through the crystallographic threefold axis (Figure 3a). The structure of ctATC2Cys is highly similar to that reported for ctATC-WT, as indicated by the low root-mean-square-deviation for the superposition of all Cα atoms in the subunit (rmsd = 0.439 Å) or in the protein trimer (rmsd = 0.704 Å) (Appendix A). The high-resolution electron density map was unambiguous and allowed for the building of the entire polypeptide chain, except for residues 2026–2032 that appeared flexibly disordered in the ctATC2Cys apo structure. These missing residues correspond to a loop (the CP-loop) that projects from one subunit into the active site of the adjacent subunit and interacts with the substrate [18] (Figure 3a). Indeed, similar ctATC2Cys crystals grown with CP (Table 1) showed extra electron density at the active site assigned to the bound substrate and residues S2029 and K2032 from the CP-loop, which was well-defined in this structure (Figure 3b). The binding of CP to ctATC2Cys was as already described for ctATC-WT [18], including a small hinge closure of the protein subunit (Appendix A).

The only important difference in the structures of ctATC2Cys that are free or bound to CP was the presence of the mutated Cys residues and the extra electron density between their side chains that demonstrates the formation of three disulfide bridges between subunits in the trimer (Figure 3c). Thus, we conclude that the Cys and formation of the disulfide bridges do not alter the protein folding or oligomerization and do not interfere with the binding of CP. However, we could not grow crystals of ctATC2Cys in complex with PALA (N-phosphonoacetyl-L-aspartate), a transition state analog that induces a large conformational change of the Asp-domain that is required for the reaction (Appendix A) [21,22,23,29]. Therefore, it was not clear whether the Cys bridge could be preventing larger conformational movements and hampering protein activity.

### 2.3. Increased Stability of the ctATC2Cys Trimer

To study the effect of disulfide bridge formation, we compared the stability of ctATC-WT and ctATC2Cys by differential scanning fluorimetry (DSF). The denaturation curves showed that ctATC-WT had a melting temperature (T_m_) of 53.9 °C (Figure 4a), similar to that of the isolated ATC domain of human CAD [23]. Instead, the double mutant ctATC2Cys had a T_m_ = 73.5 °C, 20 °C higher than ctATC-WT, and similar to the T_m_ of human ATC bound to PALA, which induces a tight and nearly irreversible complex [23]. Both for ctATC-WT and ctATC2Cys, the stability increased in the phosphate buffer (T_m_ = 70.4 °C and 80.3 °C, respectively) (Figure 4a), likely due to the binding of inorganic phosphate to the CP binding site, as reported for human ATC [23]. However, more importantly, although we did not obtain crystals with PALA, the addition of this compound increased the T_m_ of ctATC-WT and ctATC2Cys to ~90 °C (Figure 4b), confirming that the transition state analog binds and most probably induces the closure of the subunits [23].

To confirm that the increased stability of ctATC2Cys was due to the formation of the disulfide bridges, we examined the stability of the protein at increasing concentrations of the reducing agents β-mercaptoethanol (βME) or dithiothreitol (DTT) (Figure 4c,d). These assays showed that while the stability of ctATC-WT was not affected, the stability of ctATC2Cys decreased exponentially with the increasing concentration of either reducing agent. Extrapolation of the curves to an infinite concentration of the chemical indicated that Cys substitutions increase the T_m_ by 3–4 °C and that the remainder is due to disulfide bridge formation (Figure 4e).

### 2.4. Cysteine Bridges Do Not Hinder Enzymatic Activity

We compared the enzymatic activity of ctATC-WT and ctATC2Cys using a colorimetric assay that quantifies the production of carbamoyl aspartate (Figure 5a). In reactions containing 5 mM CP and 10 mM Asp, ctATC-WT exhibited a velocity v = 3.53 ± 1.45 mmoles·h^−1^·mg^−1^, which is similar to that observed for the isolated ATC domain of human CAD (v = 3.76 ± 0.23 mmoles·h^−1^·mg^−1^) [23] and is also comparable to the activity of the *C. thermophilum* construct spanning the ATC and DHO-like domains (v = 4.08 mmoles·h^−1^·mg^−1^ of ATC; considering that the ATC domain is ~42% of the DHO-ATC construct) [18]. On the other hand, the activity assays with ctATC2Cys showed that the mutations to Cys increased the activity of the enzyme, making it two-fold faster than the ctATC-WT (v = 8.87 ± 1.06 mmoles·h^−1^·mg^−1^) (Figure 5a). The Asp-saturation assays (at 5 mM CP) showed that despite this higher activity, the kinetic curves of ctATC2Cys were similar to ctATC-WT, with an S_0.5_^Asp^ = 1.5–2 mM and a pronounced inhibition at high concentrations of Asp (Figure 5b). 

Overall, these assays indicated that disulfide bond formation, rather than hampering protein activity, increases the specific activity, most likely by preventing dissociation of the active trimer at the high dilutions used for the assay.

## 3. Discussion

The large size and symmetry of the multi-enzymatic protein CAD should make it an ideal target for structural studies by cryo-EM, even more considering the advances of this technique in recent years [30]. However, attempts by many groups—including ours—to produce stable mammalian CAD particles have been frustrated by the large size and sensitivity to proteolytic cleavage of the isolated particles. In search of a more stable protein, we turned our attention to the CAD-like protein from the thermophilic fungus *C. thermophilum*, as the structural similarity between the ATC and DHO-like domains with those of the human protein suggested that both proteins share a common architecture [18]. Thus, although not published, we managed to express the recombinant full-length ctCAD-like protein and to isolate the hexameric particles, but unfortunately, the characterization by cryo-EM was hampered by the dissociation of the particles in the EM grids, probably due to the low concentrations used for vitrification and also to destructive forces at the air–water interface [31]. Similar problems in other protein complexes were solved by various approaches, including the use of crosslinking agents to stabilize protein interactions, although these may have the disadvantage of affecting protein structure or activity [32] or forming unwanted aggregates, as we experienced with ctCAD-like samples. Alternatively, due to the low natural abundance, cysteine residues can be engineered to favor disulfide bond formation and specific covalent crosslinking of protein oligomers or complexes [33], as reported in pioneering studies of phage T4 lysozyme [34,35] and, more recently, in the characterization of the SARS-CoV-2 Spike protein [36]. However, this approach requires prior structural knowledge of the target protein and careful selection of the Cys positions.

The key role of ATC in the molecular organization of CAD was first demonstrated by studies from the group of D. Davidson, who proved that a point mutation prevented the trimerization of this domain and caused the dissociation of CAD hexamers [26,37]. Going the other way around, we proposed that the specific crosslinking of the ATC subunits could result in a more stable CAD particle that is less likely to dissociate. For this, and based on the crystal structure of ctATC [18], we replaced two positions with Cys, N2045, and R2238 to favor the formation of a head-to-tail covalent linkage of the subunits in the trimer. In this work, we demonstrated that the double mutation did not affect the solubility of the protein and favored the formation of disulfide bridges that facilitated the rapid crystallization of the protein (Figure 3a and Appendix A). The high-resolution crystal structures of ctATC2Cys also indicate that the disulfide bridge does not alter protein folding and neither impedes the binding of CP nor the induced conformational changes (Figure 3b and Appendix A). Moreover, the mutations to Cys per se caused an increment of 3–4 °C in the resistance of the protein to thermal denaturation, and the stability further increased by 20 °C by the formation of the disulfide bridges (Figure 4a). Thermal stability assays also proved that the disulfide bridges do not impede the binding of PALA (Figure 4b), as shown by the increase of T_m_ to nearly the detection limit upon adding this compound. Based on these results, it was not unexpected that ctATC2Cys preserves the enzymatic activity, but it is indeed surprising that such activity was two-fold higher than the WT protein (Figure 5a,b). In the absence of performing the activity assays under reducing conditions, the most likely explanation is that the disulfide bridges prevent the dissociation of the trimer at the diluted concentration used in the assay and thus, increase the specific activity of the mutant.

In summary, the present results indicate that the mutations N2045C and R2238C favor the covalent crosslinking of the ctATC domain and, therefore, could increase the stability of the ctCAD-like particles, helping to prevent their dissociation during the preparation of the samples for cryo-EM studies. Moreover, based on the structural similarity of the ATC domains between ctCAD-like and human CAD proteins, we propose that this crosslinking strategy might also be successful in stabilizing mammalian CAD. For this, only the replacement with Cys of residue E2208 (human CAD numbering) should be sufficient for the formation of the disulfide bridge since the second position (C2016 in human CAD) is already a Cys in mammalian CAD proteins. The complexity of this megaenzyme is yet to be discovered, and we do not rule out that other protein engineering tricks may be necessary to reveal its structure.

## 4. Materials and Methods

### 4.1. Cloning and Site-Directed Mutagenesis

The *Chaetomium thermophilum* CAD-like (ctCAD) synthetic gene was purchased from GenScript based on the sequence retrieved from the genome resource (http://ct.bork.embl.de, accessed on 28 July 2015). The ctATC encoding fragment was amplified by PCR using Phusion High-Fidelity DNA Polymerase (New England Biolabs, Ipswich, MA, USA) and a pair of oligonucleotides (Appendix A) with flanking regions for In-Fusion cloning in pOPIN-M vector (Oxford Protein Production Facility) [38] linearized with restriction enzymes HindIII and KpnI. The resulting construct pOPIN-M-ctATC-WT was verified by Sanger sequencing and encoded the protein fused at the N-terminus to a histidine-tagged maltose binding protein (His_6_-MBP) with a PreScission cleavage site. Mutations N2045C and R2238C were introduced in the plasmid by overlap extension PCR [39] using flanking and mutagenic primers (Appendix A). The amplified PCR product with the double mutation was cloned in pOPIN-M as described above, and the resulting construct, pOPIN-M-ctATC2Cys, was verified by Sanger sequencing.

### 4.2. Protein Expression

The ctATC-WT and ctATC2Cys proteins were expressed in *E. coli* BL21 (DE3) pLySs cells (Merck, Darmstadt, Germany) using auto-induction media [40]. Shortly, a preculture of transformed *E. coli* BL21 (DE3) pLysS was grown overnight at 37 °C in 50 mL of LB medium supplemented with ampicillin (100 μg·ml^−1^), chloramphenicol (34 μg·ml^−1^) and 2% (*w*/*v*) glucose. The preculture was added to 1 L of ZY medium (1% (*w*/*v*) tryptone and 0.5% (*w*/*v*) yeast extract) supplemented with 1 mM MgSO_4_, 1 × 5052 solution (0.5% (*v*/*v*) glycerol, 0.05% (*w*/*v*) glucose, and 0.2% (*w*/*v*) α-lactose) and 1 × NPS solution (50 mM Na_2_HPO_4_, 50 mM KH_2_PO_4_, and 25 mM (NH_4_)_2_SO_4_), and ampicillin (100 μg·ml^−1^) and chloramphenicol (34 μg·ml^−1^). After 6 h incubation at 37 °C in a shaker at 200 rpm, the temperature was decreased to 20 °C for overnight growth. The cells were harvested by centrifugation, washed with phosphate buffer saline, and stored at −80 °C.

### 4.3. Protein Purification

Cells were resuspended in buffer A (20 mM Tris-HCl pH 8.0, 500 mM NaCl, 5% (*w*/*v*) glycerol, 10 mM imidazole) supplemented with 1 mM Pefabloc (Merck) and lysed by sonication. Following clarification by centrifugation at 35,000× *g* for 45 min, the supernatant was filtered and loaded into a 5 mL Nickel-NTA agarose cartridge (ABT) equilibrated in buffer A. After extensive washing with buffer A containing 35 mM imidazole, the protein was eluted by increasing the imidazole concentration to 250 mM. The eluted protein was dialyzed overnight against buffer A with 35 mM imidazole and digested with GST-PreScission protease, which was added to the dialysis bag at 1/20th of the protein weight to cleave the His_6_-MBP tag. Then, the sample was loaded into a Nickel-NTA agarose cartridge attached to a 5 mL GST-Trap HP (GE) column equilibrated in buffer A with 35 mM imidazole. The untagged protein eluted in the non-bound fraction and was concentrated in an Amicon Ultra centrifugal filter unit (Merck) with a 30 kDa cutoff to ~3 mg·ml^−1^. The sample was further purified through size exclusion chromatography (SEC) Superdex 200 10/300 GL (GE) column equilibrated in SEC buffer (20 mM Tris-HCl pH 8.0 and 150 mM NaCl). All purification steps were carried out at 4 °C except SEC, which was carried out at room temperature. The purified protein was supplemented with 20% glycerol, frozen in liquid nitrogen, and stored at −80 °C.

### 4.4. Crystallization

Initial crystallization screenings were performed at 18 °C in MRC crystallization plates (Molecular Dimensions, Rotherham, UK) with drops of 0.2 μL protein at 3 mg·mL^−1^ with or without 2 mM CP (Merck) in SEC buffer and 0.2 µL reservoir solution from the commercial screenings JCSG+ and PACT (Qiagen, Hilden, Germany). The protein without CP formed bipyramidal-shaped crystals in less than 20 min, and conditions were optimized in hanging drop 24-well plates (Hampton Research) mixing 1.5 µL of protein at 3 mg·mL^−1^ and 1.5 µL of reservoir solution. Best crystals appeared using as precipitant 0.8 M sodium succinate at pH 7.0. Crystals containing CP grew in conditions with 0.1 M Tris-HCl pH 8.5, 0.1 M NaCl and 0.6 M ammonium acetate. Cryo-protection was reached by directly soaking the crystals in mother liquor supplemented with 20% of glycerol and flash-cooled in liquid nitrogen. 

### 4.5. Data Collection and Structure Determination

X-ray diffraction datasets were collected at XALOC BL-13 (ALBA, Barcelona, Spain) beamline using a Pilatus 6M detector and processed automatically with AutoProc [41]. Crystallographic phases were obtained by molecular replacement using PHASER [42], and the structures of the wild-type protein in apo-sate (PDB ID 5G1O) or bound to CP (5G1N) as search models. Model building and refinement were carried out by iterative cycles in COOT [43], PHENIX [44], and Refmac5 [45].

### 4.6. Differential Scanning Fluorimetry (DSF)

Protein stability was measured by differential scanning fluorometry [46] using a 7500 Real-Time PCR system (Applied Biosystems, Waltham, MA, USA) in a 96-well reaction plate with 40 µL samples with 5 µM protein, 5x SYPRO Orange (Invitrogen, Waltham, MA, USA), and 20 mM Tris-HCl pH 7.0 or sodium phosphate pH 7.0. PALA was obtained from the Developmental Therapeutics Program (NCI, National Institutes of Health, MD, USA) Open Chemical Repository and was added to the sample at 1 mM final concentration. DTT (dithiothreitol; Apollo Scientific, Cheshire, UK) or βME (β-mercaptoethanol; Merck) were added to the samples at the final concentration range of 0–50 mM. Fluorescence changes were monitored every 1°C in a temperature ramp from 20 to 95 °C using the extrinsic fluorescence of SYPRO Orange (λ_excitation_ = 465 nm, λ_emission_ = 580 nm). Curves were normalized, and the melting temperature (T_m_) was determined as the midpoint of the unfolding transition. Data were analyzed and plotted with GraphPad. 

### 4.7. Activity Assays

Enzymatic activity was measured with a colorimetric assay that quantifies the production of carbamoyl aspartate [47]. The reactions were carried out at 25 °C in a final volume of 100 µL containing 50 mM Tris-acetate pH 8.3, 0.1 mg·ml^−1^ bovine serum albumin (BSA; Merck), and 0.1 µM of ctATC-WT or ctATC2Cys. The activity was measured at fixed concentrations of CP (5 mM; Merck) and aspartate (10 mM; Merck) or varying concentrations of aspartate between 0.15 and 40 mM. The protein was incubated with aspartate for 10 min at 25 °C, and the reaction was triggered by the addition of CP and stopped at different time points by adding 0.25 mL of a color mix containing two parts of reagent A (0.37% (*w*/*v*) antipyrine and 0.25% (*w*/*v*) ammonium iron (III) sulfate in 25% H_2_SO_4_ (*v*/*v*) and 25% (*v*/*v*) H_3_PO_4_) and one part of reagent B (0.4% (*w*/*v*) diacetylmonoxime in 7.5% (*w*/*v*) NaCl). Samples were boiled at 95 °C for 15 min and cooled down in the dark for 30 min for color development. The absorbance of the samples at 465 nm was measured in a spectrophotometer Eppendorf BioSpectrometer Kinetic (Eppendorf, Hamburg, Germany). A standard color curve was made using serial dilutions of carbamoyl aspartate (Merck). Data were analyzed with GraphPad, and kinetic curves were fitted to a substrate inhibition model according to the equation v = v_max_ * S/(K_m_ + S * (1 + S/K_i_)), where v and v_max_ are the velocity and maximum velocity, respectively, S is the aspartate concentration, K_m_ is the Michaelis–Menten constant, and K_i_ is the inhibition constant.

## Figures and Tables

**Figure 1 molecules-28-00660-f001:**
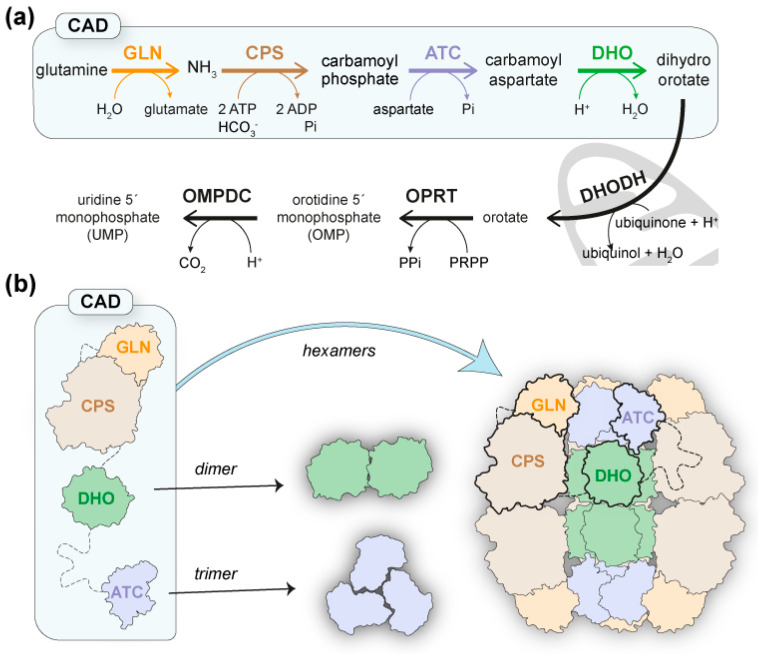
**CAD and the de novo pyrimidine biosynthesis pathway.** (**a**) Scheme of the conserved reactions and enzymatic activities for UMP synthesis: GLN, glutaminase; CPS, carbamoyl phosphate synthetase; ATC, aspartate transcarbamoylase; DHO, dihydroorotase; DHODH, dihydroorotase dehydrogenase; OPRT, orotate phosphoribosyltransferase; OMPDC, orotidine monophosphate decarboxylase. (**b**) Schematic representation of CAD with enzymatic domains in different colors and linkers shown as dashed lines. A model for the architecture of CAD is proposed where the assembly of ATC and DHO domains as trimers and dimers, respectively, nucleate the association of CAD into hexameric particles.

**Figure 2 molecules-28-00660-f002:**
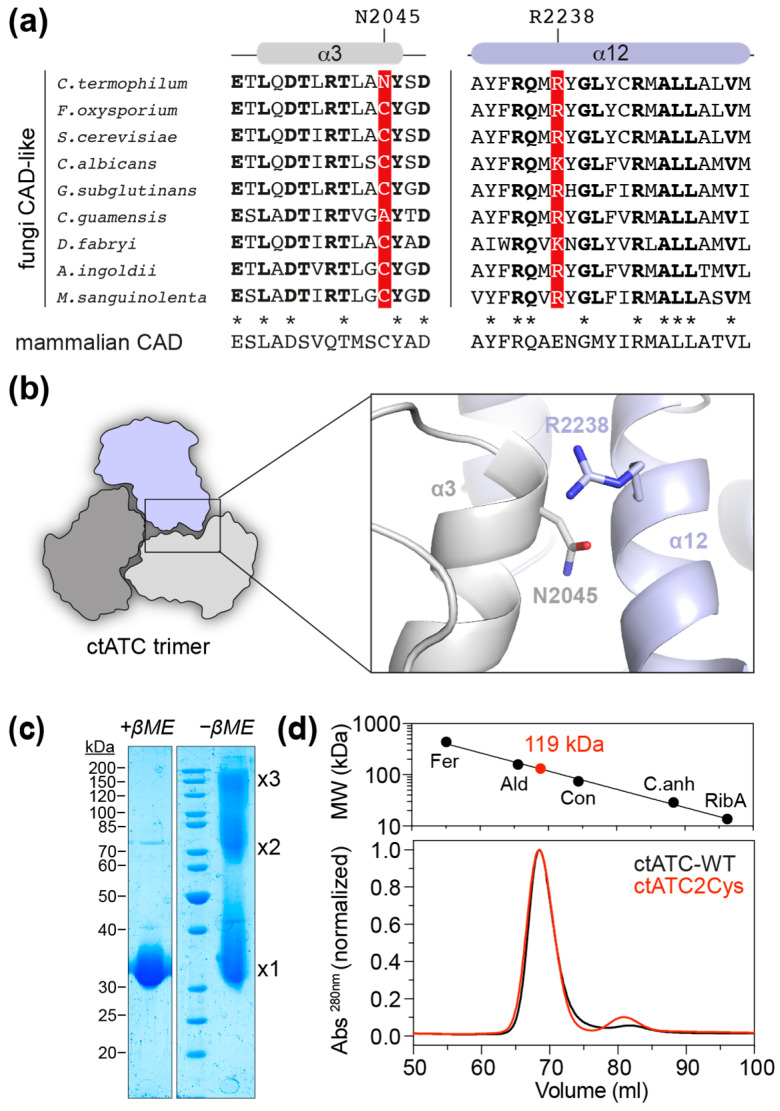
**Mutant design and purification.** (**a**) Alignment of the sequences corresponding to helices α3 and α12 in fungi CAD-like proteins: *Fusarium oxysporum* (UniProt A0A2H3T979); *Saccharomyces cerevisiae* (UniProt P07259); *Candida albicans* (Uniprot A0A1D8PTD1); *Gibberella subglutinans* (Uniprot A0A8H5Q5Q2); *Ceraceosorus guamensis* (Uniprot A0A316VUP0); *Debaryomyces fabryi* (Uniprot A0A0V1PUU2); *Acaromyces ingoldii* (Uniprot A0A316YPF6); *Mycena sanguinolenta* (Uniprot A0A8H7D3Z0). Residues in bold are identical in the aligned fungal sequences. Residues N2045 and R2238 in *C. thermophilum* ATC and the corresponding positions in the other proteins are shown in red background. The corresponding invariant sequence of mammalian CAD is shown underneath. Conserved residues in fungi and mammals are denoted with an asterisk. The position N2045 in *C. thermophilum* is a Cys in many fungal sequences and mammalian CAD. (**b**) Cartoon representation of the *C. termophilum* ATC trimer and detail of the proximity between N2045 and R2238 in the crystal structure (PDB 5NNQ). (**c**) SDS-PAGE of purified ctATC2Cys with and without β-mercaptoethanol (βME) in the loading buffer. Bands corresponding to protein monomer, dimer, and trimer are indicated. (**d**) SEC elution profile of ctATC-WT and ctATC2Cys and estimation of molecular weight based on the calibration with standards of known molecular weight (Fer, ferritin 440 kDa; Ald, aldolase 158 kDa; Con, conalbumin 75 kDa; C.anh, carbonic anhydrase 29 kDa; RibA, ribonuclease A 13.7 kDa).

**Figure 3 molecules-28-00660-f003:**
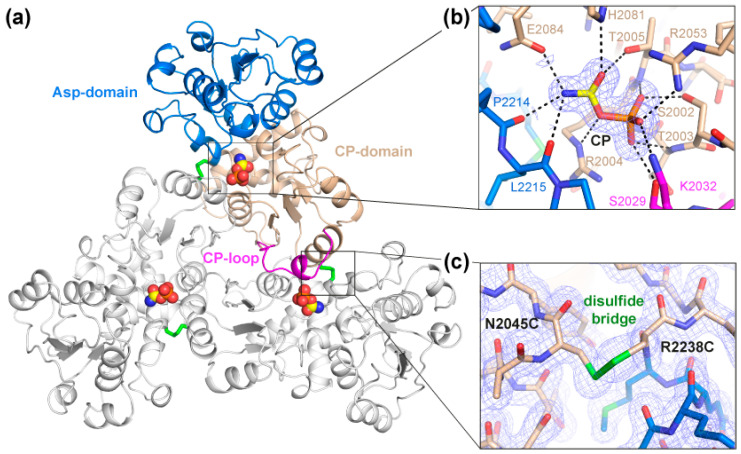
**Crystal Structure of ctATC2Cys.** (**a**) Cartoon representation of ctATC2Cys trimer bound to CP (yellow spheres). One subunit is colored with the Asp- and CP-domains in blue and brown, respectively, and the CP-loop in magenta, and the other subunits are colored white. CP is shown as spheres and disulfide bridges as green sticks. (**b**) Detailed view of CP binding. The substrate is shown within the 2F_obs_–F_calc_ electron density map represented as blue mesh. Dashed lines indicate electrostatic interactions with the ligand. (**c**) Detailed view of the mutations to Cys and formation of a disulfide bridge represented as a green bond. The 2F_obs_–F_calc_ electron density maps are contoured at 1 σ.

**Figure 4 molecules-28-00660-f004:**
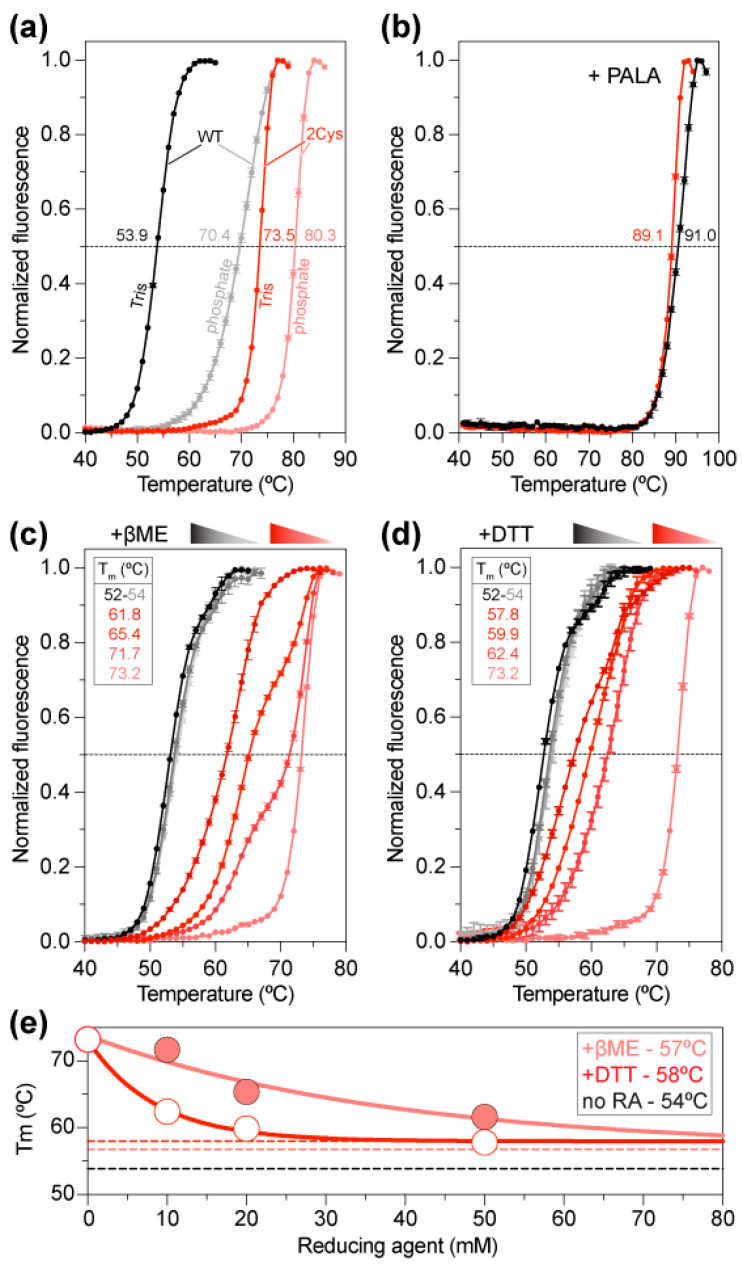
**Thermal stability of ctATC-WT and ctATC2Cys.** Thermal stability curves measured by DSF for ctATC-WT (grayish) and ctATC2Cys (reddish) in Tris or phosphate buffers (**a**), with PALA (**b**), or at varying concentrations of β-mercaptoethanol (βME) (**c**) and dithiothreitol (DTT) (**d**). The melting temperatures (T_m_) that occur at the midpoint of the unfolding transitions are indicated. (**e**) T_m_ exponential decay at increasing concentrations of the reducing agents. The dashed lines indicate the T_m_ in absence of a reducing agent (no RA, black) or at infinite concentration of βME (salmon) or DTT (red). Errors bars indicate the standard deviation of three independent measurements.

**Figure 5 molecules-28-00660-f005:**
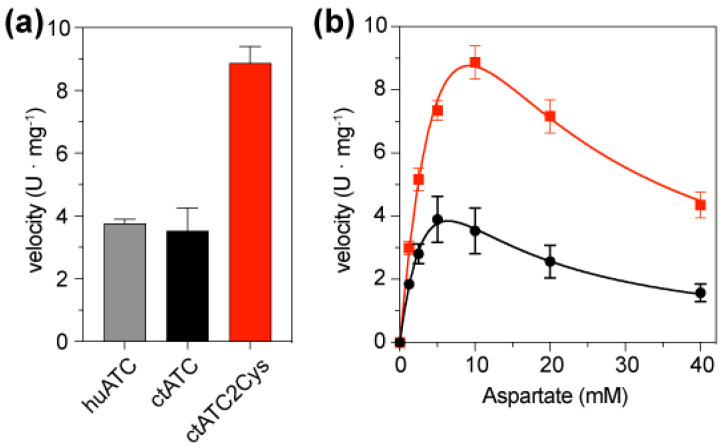
**Enzymatic activity of ctATC-WT and ctATC2Cys.** (**a**) Bar graph of the enzymatic velocity of human ATC (grey), ctATC-WT (black), and ctATC2Cys (red) measured at 5 mM CP and 10 mM Asp. (**b**) Activity assay at a fixed concentration of CP (5 mM) and varying the concentration of Asp. The curves show the fitting to a substrate inhibition model. Errors bars indicate the standard deviation of three independent measurements. One enzyme unit (U) is the amount of enzyme producing 1 mmol of carbamoyl aspartate per hour.

**Table 1 molecules-28-00660-t001:** Data collection and refinement statistics for ctATC2Cys structures. Statistics for the highest-resolution shell are shown in parentheses.

	Apo-State	Bound to CP
**Data Collection**		
Wavelength (Å)	0.97926	0.97926
Space group	P4_3_32	P4_3_32
Unit cell: a, b, c (Å) α, β, γ (°)	139.8, 139.8, 139.8 90, 90, 90	139.4, 139.4, 139.4 90, 90, 90
Resolution (Å)	49.08–2.03 (2.10–2.03)	49.29–1.58 (1.64–1.58)
Reflections (observed/unique)	325,801/30,158 (30,416/2906)	900,158/63,482 (78,877/5919)
Multiplicity	10.8 (10.5)	14.2 (13.3)
R_pim_	0.019 (0.43)	0.021 (0.71)
I/σ_I_	19.9 (1.7)	19.8 (1.1)
Completeness (%)	100 (100)	99.7 (96.8)
CC_1/2_	1.0 (0.70)	1.0 (0.45)
Wilson B factor (Å^2^)	47.67	26.57
**Refinement**		
Resolution (Å)	49.08–2.03	49.29–1.58
Reflections	30,113	63,194
R-factor/R_free_ (%)	17.11/19.99	14.58/17.32
R.m.s. deviations		
Bond lengths (Å)	0.006	0.013
Bond angles (°)	0.864	1.21
N° atoms (no H)		
Protein + ligand	2594	2644
Water	62	291
Ramachandran plot		
Favored (%)	96.74	97.78
Allowed (%)	3.26	2.22
Outliers (%)	0.00	0.00

## Data Availability

The coordinates and structure factors for ctATC2Cys free and bound to CP were deposited in the Protein Data Bank under accession codes 8BPS and 8BPL, respectively.

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
