# Peer review of "A Tailored Strategy to Crosslink the Aspartate Transcarbamoylase Domain of the Multienzymatic Protein CAD"

_molecules, 2023, doi:10.3390/molecules28020660_

Round 1

Reviewer 1 Report

Title: “A tailored strategy to crosslink the aspartate transcarbamoylase domain of the multienzymatic protein CAD “

Authors: Francisco Del Cano-Ochoa , Antonio Rubio-del-Campo , Santiago Ramón-Maiques  

CAD is a key multidomain protein in de novo pyrimidine biosynthesis, which, in addition to be a potential target for the development of antitumor drugs, is involved in an infantile genetic disorder. Its great interest justifies the numerous attempts to understand this 1.5MDa megaprotein and to determine its structure. However, its complexity and size have led to great difficulties in the production and characterization of this protein.

In order to solve some of the limitations that imply working with human CAD, the authors use the CAD-like from C. thermophilum as an homologous model, having generated several constructs encoding for some of the isolated domains. In this manuscript del Caño et al. propose and test as a strategy to stabilize the core of CAD and prevent its dissociation, the generation of covalent interactions between the ATC domains from different subunits.

 Using site-directed mutagenesis, the authors generate two mutations to cysteine that lead to the formation of disulfide bridges between ATC subunits and discard that the presence of these mutation generate folding artifacts or prevents catalytic activity or substrate binding. On the other hand, they test the stabilizing effect of the presence of these mutations, mainly due to the covalent binding of the subunits by disulfide bridges. These results open the doors to trying this strategy as a mechanism that stabilizes CAD, avoiding its dissociation and facilitating structural studies with this protein.

The objective of the work is coherent, the strategy seems correct and the experiments are appropriate. In fact, the results represent an advance in the direction of stabilizing the protein, which is a requirement to determine its structure, a goal to understand the function of CAD. For all these reasons, I consider the publication of this work pertinent.   However, I have some considerations to it and some minor comments or suggestions:   1-On the one hand, the authors indicate that although the CAD-like of C. thermophilum does not present the degradation problems of the human CAD, it fell apart during the preparation of grids for EM (lines 81-84). It is not indicated if there is any other evidence of the dissociation of the sample. On the other hand, previous works trying to stabilize the core of this protein through crosslinking, resulted in heterogeneous samples that were not suitable for structural analysis. Given the stabilization obtained in this work by covalently binding of  the ATC domains, and the fact that generation of disulfide bridges is a more defined way to bind convalently the different subunits that crosslinking, it would be good to test whether the present strategy is effective in overcoming the previous problems with more complex constructions (DHO-ATC or CAD-like). I suggest to the authors that, if available, they provide any evidence they have supporting that the stabilization strategy that they prove to be effective in the present work, is also effective to stabilize DHO-ATC or the complete CAD protein. This would give more value to the present work.

2- The authors try to generate disulfide bridges by introducing cysteine ​​mutations in specific positions of the protein and they carry it out successfully. I wonder, since it does not indicate it in the text, if they tried several combinations of mutants for this purpose or if, on the contrary, they were right the first time in the combination of appropriate mutations that gave rise to the formation of disulfide bridges without affecting folding or of the enzymatic activity.

3- Writing of results section corresponding to the description of the crystallographic structure should be reviewed. On the one hand, the main objective of this section is not to describe the structure itself but to compare it with the previous structure of the non-mutated protein to describe the differences, given that the main purpose is to analyze the possible consequences of the presence of the mutation on the folding or on the active center. In this direction, I think the structures being compared in each sentence should be clearer: lines 155-156, I understand that it refers to the changes observed by the union of CP either in the WT structure or in the mutant; lines 166-169, refers to the comparison between either the ligand-free, or CP-bound structures, between the wild form and the mutant form. On the other hand, just as the rmsd value is given for subunit superimposition, the rmsd value could be also given for trimer superimposition and comment on whether or not disulfide bond formation affects oligomer conformation. Given the greater activity of the mutant protein, it could be analyzed whether the structure of the mutant free from ligands, presents some trait as the stabilization of some catalytic elements or even a conformation of the trimer more similar to that for the catalitically active protein, that could explain its higher activity.

4-In relation to the thermal stability experiments and specifically with figure 4, I have several comments and suggestions: Both in material and methods and in the figures, it is indicated that the data have been normalized, however in the figure it is not clear how this normalization has been done, because for some curves, it seems that the saturation is reached at values ​​lower than 1 while for others, values ​​greater than 1 are represented and no saturation is even observed in the curves. Lines 354-355. “Tm was determined as the midpoint of the unfolding transition. Data were analyzed and plotted with GraphPad”. I believe that the entire range of data used in determining the Tm should be represented. It is also worth noting the nature of the curves at 10 and 20 mM of BME, since they seem to present double sigmoidal nature. It should be indicated the way in which the curves have been analyzed to obtain the Tm values. On the other hand, I recognize the difficulty of representing and identifying curves corresponding to different concentrations in the same panel, but I think it is necessary to avoid that the identification of the curves (mM of BME or DTT) as well as the Tm corresponding to each one, remain above the curve. I suggest trying a wider format for panels c and d of figure 4 or using a color code with legend to indicate the different concentrations and taking the Tm values ​​to the side or as an inset (if there is space for it).  The denaturation curves represented should correspond to duplicates or triplicates, and this should be indicated in material and methods or in the figure legend. In any case, the values ​​of Tm represented in panel e should include standard deviation or standard error.

5- In the section corresponding to activity assays, the authors compare the activities of the wild-type and mutant protein as well as with the human protein. According to what is indicated in the text, the specific activity values ​​provided correspond to determined concentrations of both substrates and therefore should not be given as Vmax but as v. When the activity of the protein that includes the ATC and DHO domains is given, it is not clear to me if the specific activity refers to mg of complete protein or to the fraction corresponding to the ATC domain. In case that it correspond to the complete protein, the activity values ​​would not be comparable to those obtained for the isolated domain. Please clarify in the text.   6-The activity results discard a negative effect of the mutations on the activity, since, on the contrary, the activity for the mutant is 2-fold higher. Among the possible causes for this, the authors discard that it is due to changes in the kinetics for aspartate.The authors suggest that there might be dissociation of the wt protein at the low concentrations used in the assay, which could be prevented by introducing disulfide bonds into the mutant protein. If so, I was wondering if it would be possible to carry out an assay at variable enzyme concentrations, to see if greater specific activity for the wild protein was obtained at higher protein concentrations. This would support the mechanism suggested by the authors. On the other hand, the increased activity of the mutant protein could be due to the stabilization of the active conformation. Given that the active center of the enzyme is at the interface between subunits, I believe that, as I have commented previously, it would be convenient to carry out an superimposition analysis of the trimers to determine if the the mutant protein in the absence of substrates could could present a conformation catalytically more favorable that the wild type form.

Throughout the text, I find other minor aspects that I consider could be reviewed:   Line 19. The term “covalent oligomerization” is not very common. Authors could consider to use “formation of covalent oligomers” instead.   Lines 29 and 30. Please change “synthesis” by “biosynthesis”   Lines 65-68. In the introduction, the authors indicate that although the DHO domain is inactive in C. termophilum CAD-like, it forms dimers as in Human CAD, so it is possible that it plays the same structuring role in protein architecture previously demonstrated for the HuCAD ATC domain. Sentence in lines 67-68 should be reformulated to replace “indicating” with “supporting” and “DHO plays” with “DHO could play”.   Figure 2. The lower panel of Figure 2d shows absorbance versus elution volume. I understand that absorbance values ​​are normalized, which should be indicated in the name of the ordinate axis.   Line 113- “that the cys did not impair” is imprecise. Specify the change, reformulating the sentence: ”that a cys at position xxx did not impair…” or “that mutation xxx did not impair…” In the same sentence, line 113, change "decrease" by "decreased"   Line 189-190. The authors suggest that the shift of the denaturation curves in the presence of PALA, which supports the binding of this inhibitor, confirms that PALA binding “most probably induces subunit closure”. This statement is based on prior structural knowledge, which should be referred to in this sentence, since the statement is made by analogy to the conformational changes described when comparing structures without/with PALAreported in other works.   Line 287. I would change the use of "code" to "encode" as a verb when referring to the genetic code, because I think it is more appropriate. In fact, this is how it is done in the introduction (lines 33 and 37)   line 297 “…constructs were expressed”. Proteins are expressed from the constructs.   line 299 “a preculture of transformed in E. coli BL21”I think it is not correctly expressed
Figure 5- The nomenclature of the axes should include the magnitude represented and not only its units. “CASP” should be defined in the figure legend. Alternatively, the unit of enzymatic activity could be defined in the text, and the activity indicated in the figure as Umg-1
  The protein is expressed from an expression vector that includes in the produced protein a maltose binding protein tag. The work describes vector preparation and protein production.During protein purification the tag is cleaved and separated, as it is well illustrated in figure S1, however I think that the presence of this tag in the overexpressed protein could be mentioned in the text, since it is probably an important stabilizing factor for its proper overexpression in bacteria.   The thermal stability assays discriminate between two stabilizing effects: the mutation to cysteine ​​itself, and the formation of disulfide bridges. The authors verified that the second of these components, the formation of covalent interactions, is the one with the greatest contribution to thermal stabilization, which supports that this strategy could be effective in stabilizing the CAD oligomer and preventing its dissociation. However, CAD is a very complex protein with different domains and although ATC and DHO seem to play a central role in the architecture, it cannot be discarded that strengthening of ATC subunit interactions could not be enough to prevent CAD dissociation. In the absence of evidence that this strategy solves the dissociation problem with the complete protein protein, the uncertainty about the effect of this strategy on the stabilization of CAD and possible alternatives could be mentioned in the discussion.

Author Response

Please find in the attached document our replies to the comments from Reviewer #1

Reviewer 2 Report

The manuscript « A tailored strategy to crosslink the aspartate trancscarbamoylase domain of the multienzymatic protein CAD» submitted by Fraccisco del Cano-Ochoa et al. is devoted to the

specific crosslinking strategy to enhance the stability of CAD.

Authors described this mega-enzymes and the problems with the determination of its structural characterization. They proposed to introduce two cysteine residues in proteins to form additional disulfide bridge between subunits. The assays proved that disulfide bond formation increase thermal stability of the complex, prevent dissociation of trimer and increase the specific activity.

I think this work will be interesting for a wide range of researcher working on molecular biology and bioorganic chemistry.

Author Response

Please find in the attached document our replies to the comments from Reviewer #2

Reviewer 3 Report

The presented work is sound and interesting, the authors report a mutation of specific aminoacids by cysteines strategy. The authors proved that these changes stabilized the ATC domain with no effect on enzymatic activity. I missed some basic molecular dynamics simulations which is very easy to do to complement the work and verify what computational methods may say about complex stabilization.

Author Response

Please find in the attached document our replies to the comments from Reviewer #3
